# Maternal interoceptive focus is associated with greater reported engagement in mother-infant stroking and rocking

**Rosie Donaghy** [1]*, **Jeanne Shinskey**[1], **Manos Tsakiris**[1,2]

**1** Department of Psychology, Royal Holloway, University of London, Egham, United Kingdom, **2** Centre for the Politics of Feelings, Senate House, School of Advanced Study, University of London, Egham, London, United Kingdom

These authors contributed equally to this work.
\* rosie.donaghy@rhul.ac.uk

**Data Availability Statement:** The data that support these studies' findings are openly available in OSF at https://osf.io/hgpsa/.

**Funding:** MT was supported by the European Research Council Consolidator Grant (ERC-2016-

## Abstract

Parental caregiving during infancy is primarily aimed at the regulation of infants' physiological and emotional states. Recent models of embodied cognition propose that interoception, i.e., the perception of internal bodily states, may influence the quality and quantity of parent-infant caregiving. Yet, empirical investigations into this relationship remain scarce. Across two online studies of mothers with 6- to 18-month-old infants during Covid-19 lockdowns, we examined whether mothers' self-reported engagement in stroking and rocking their infant was related to self-reported interoceptive abilities. Additional measures included retrospective accounts of pregnancy and postnatal body satisfaction, and mothers' reports of their infant's understanding of vocabulary relating to body parts. In Study 1 ($N = 151$) and Study 2 ($N = 111$), mothers reported their engagement in caregiving behaviours and their tendency to focus on and regulate bodily states. In a subsample from Study 2 ($N = 49$), we also obtained an objective measure of cardiac interoceptive accuracy using an online heartbeat counting task. Across both studies, the tendency to focus on and regulate interoceptive states was associated with greater mother-infant stroking and rocking. Conversely, we found no evidence for a relationship between objective interoceptive accuracy and caregiving. The findings suggest that interoception may play a role in parental engagement in stroking and rocking, however, in-person dyadic studies are warranted to further investigate this relationship.

## Introduction

Human infants depend upon the bodily and affective provisions provided by the primary caregiver [1–4] and as such, parental care is critical for healthy infant development [5]. Importantly, the capacity to engage in sensitive caregiving is affected by the caregiver's ability to make inferences about their infant's current mental states [6, 7], yet, the mechanisms that enable these inferences are not well understood. Recent accounts have emphasised that the ability to represent and relate to others' states may be facilitated by interoception [1, 8], that is,

CoG-724537) for the INtheSELF project under the HORIZON2020 program. Manos Tsakiris: 0000-0001-7753-7576. https://erc.europa.eu/ The funders had no role in study design, data collection and analysis, decision to publish, or preparation of the manuscript.

**Competing interests:** The authors have declared that no competing interests exist.

the capacity to sense, interpret and manage changes in bodily sensations arising from within one's own body [9, 10]. Processing one's internal bodily states may provide a mechanism that facilitates mother-infant caregiving, as awareness of one's own states may translate into better awareness of someone else's bodily states. The two studies presented here aimed to explore whether variation in maternal interoception related to mothers' engagement in caregiving behaviours with their 6- to 18-month-old infants.

Interoception is a multidimensional construct that can be measured across distinct levels of perception and behaviour [11, 12]. Interoceptive accuracy quantifies an individual's objective ability to detect interoceptive states, e.g., heartbeats, using behavioural tasks [13] and it is possible to measure *beliefs* about interoceptive accuracy using self-report questionnaires [12]. Other self-report measures, such as the Multidimensional Assessment of Interoceptive Awareness (MAIA-2) [14] trace an individual's tendency to focus on and attend to bodily states [10]. Whilst the latter does not capture how accurately an individual can perceive interoceptive sensations, these self-reports provide valuable insights into people's variability in attending to and managing these states [15].

A body of evidence suggests that interoception informs our experience and regulation of emotional states [16, 17] and is involved in processes that influence how we relate and respond to others, such as empathy [18] and attachment [19]. Poorer self-reported interoceptive abilities (as reported in the MAIA-2) have been associated with deficits in emotion regulation in healthy populations [20] which contributed to depressive symptomatology [21]. Importantly, interoception allows us to monitor how our internal bodily states may be affected by the states of others [1, 8, 22, 23]. For example, direct eye contact and human touch increase the subjective experience of interoceptive sensations and modulate physiological responses to emotional stimuli [24–27].

Given the primacy of emotion co-regulation between caregivers and infants [28, 29], the role of interoceptive processes in early caregiving may be particularly important. While empirical research in this area is scarce, theoretical accounts of emotion regulation in parenthood suggest that parents attend to their own physiological and emotional experience before selecting the response to manage their infant's needs [30, 31]. Infant distress triggers physiological changes in parents, such as increased heart rate [32] and milk flow from the breast [33]. This adaptive physiological response initiates approach-oriented and alert responses to react to infant distress [34] and has been shown to predict more sensitive caregiving in mother-infant dyads [35]. Neurobiological and hormonal changes in the transition to parenthood are thought to scaffold behavioural responses to infant cues [31]. For instance, parents' elevated levels of oxytocin postpartum enhances neural responding to infant-related stimuli [36] and greater neural responding in emotion processing circuits has been found in mothers who more closely coordinated their caregiving to their infant's signals [37]. These neural circuits include the anterior insular-cingulate cortex, an area associated with integrating interoceptive and emotional states [38, 39].

In light of this, developmental models have proposed that interoception enables parents to monitor their internal bodily signals in parallel to sensing changes in their infant's own arousal states [40]. Insights from thermal imaging studies have shown that when mothers observed their child in distress, the degree and timing of skin conductance responses closely aligned with their child's [41, 42], highlighting that the autonomic nervous system provides information to parents about their infant's arousal state, as well as their own. Importantly, mothers engage in a range of caregiving behaviours which differentially modulate infants' physiological and emotional states [6, 43–46]. Stimulating behaviours, e.g., tickling and poking, have been found to elicit negative facial expressions in infants while gentle stroking elicited more infant gazing, smiling and vocalisations [47]. Gentle stroking and skin-to-skin contact have since

been found to soothe infants by stabilising cardio-respiratory rate and regulating body temperature [48–50]. Changes in cardio-respiratory markers of arousal have also been found in 4- to 12-month-olds and their mothers during hugging, which were not found during mother-infant holding [51], highlighting the distinction between different types of caregiving behaviours and their functions on the dyad. In signalling how caregiving behaviours elicit physiological and emotional changes within the dyad, the perception and interpretation of interoceptive bodily signals may support parents in regulating theirs and their infant's arousal. In childhood, evidence suggests that the way parents verbally communicate their interoceptive states may shape how children identify and appraise their own bodily sensations [52]. Given that the words for body parts are among the first words infants learn [53], the caregiving environment may shape children's identification and labelling of their body and its parts even earlier.

Despite a growing interest in the role of interoception in caregiver-infant relations [54], there have been few empirical studies investigating how parental interoception may relate to caregiving behaviours. In one study, parents who showed greater anterior insula activity, whilst watching an interaction with their own 11-month-old vs an unfamiliar infant, engaged in more gaze, vocalisation and affectionate touch during an interaction when the child was 4 years old [55]. Other studies have found natural variation in the speed of caregiver-infant stroking [56, 57] which correlated with the speed of the mother's own heart rate [58] suggesting that qualities of mother-infant stroking may be modulated by the mother's own interoceptive system. This yields important implications because the effect of stroking in regulating infants' physiological arousal, such as heart rate, is based on experimental studies where affective touch is delivered at a velocity of 3cm/s [59, 60]. While the aforementioned studies provide evidence that interoception may be involved in the quality and quantity of mother-infant caregiving behaviours, to our knowledge, no studies have investigated whether caregiving is related to the caregiver's levels of interoception such as interoceptive accuracy or subjective appraisals of interoceptive states, i.e., their tendency to focus on interoceptive states.

Moreover, evidence suggests that interoception plays a key role in shaping body image [61–63], defined as the conscious, mostly visual, mental representation of an individual's body appearance, including its perceptual aspects (e.g., its shape, size) and its affective-cognitive evaluation (e.g., feeling dis/satisfied with the appearance or size of the body) [64]. For individuals with have poorer interoceptive abilities, their body representation seems to be more heavily influenced by exteroceptive (i.e., visual) features of their body, which may also explain why they are more susceptible to body-image disturbances [65]. Accordingly, lower interoceptive accuracy is associated with greater body dissatisfaction [66]. The substantial bodily changes experienced during pregnancy and the postpartum suggest these may be sensitive periods for interoception and body image, but few studies have investigated this. In a sample of pregnant women, lower self-reported interoceptive focus was associated with greater body dissatisfaction during pregnancy [67], which has been linked to less engagement in breastfeeding [68], highlighting that interoception and body image have the potential to influence parent-infant caregiving.

We present two studies designed to explore whether variation in maternal interoception relates to mothers' engagement in caregiving behaviours with their 6- to 18-month-old infants. As a result of the COVID-19 pandemic, the studies were conducted online and were primarily restricted to parental reports. Study 1 ($N$ = 151) explored whether mothers' subjective tendencies to focus on and manage interoceptive bodily signals related to their engagement in holding and stroking their infant and other affective communication behaviours. Participants also reported retrospective accounts of pregnancy body satisfaction, and attitudes to social touch, to assess possible relationships with interoception and caregiving. Additional analyses in Study 1 investigated whether mothers' engagement in caregiving behaviours related to infants' bodily

awareness as measured by their comprehension of vocabulary relating to body parts, using the UK-Communicative Development Inventory [69].

Study 2 was a replication and extension of Study 1 in a new sample of mothers ($N$ = 111), to investigate whether engagement in these caregiving behaviours related to mothers' accuracy in perceiving interoceptive (cardiac) sensations. To this aim, we obtained the same self-report measures of interoception and caregiving in Study 1, and a subsample completed an online version of the heartbeat counting task ($N$ = 52), where participants report the number of heart-beats they experience across several intervals. Participants completed measures of pregnancy, postnatal and current body satisfaction.

The studies allowed us to test several key hypotheses. Firstly, we predicted that mothers who reported greater interoceptive focus would also report greater engagement in mother-infant caregiving. Additional hypotheses included that (retrospective) body dissatisfaction during pregnancy would be negatively correlated with self-reported interoceptive focus, and that infants' comprehension of body-part vocabulary would be positively associated with maternal engagement in caregiving behaviours. In conducting Study 2, we tested whether mothers' interoceptive accuracy was associated with greater engagement in caregiving behaviours, and whether interoceptive accuracy was associated with (retrospective) body dissatisfaction across pregnancy, the postnatal period and at present.

## Study 1

### Materials and methods

**Participants.** Our target sample size was 146, which we obtained by calculating a power analysis (G*Power v3.0; Faul et al., 2007). Our goal for a fixed model regression was to obtain .95 power to detect a small effect size of .15 at the standard .05 alpha error probability.

The study was conducted in line with regulations and approved by the Institutional Ethics Committee at Royal Holloway University of London. Eligible participants in the Departmental family database were invited to take part in the online study and we also advertised the study through social media in parent groups in the surrounding areas of the University. The study was hosted on Qualtrics in May 2020 (https://www.qualtrics.com/) and all participants identified as residents in the United Kingdom. Upon giving informed written consent, participants remained free to withdraw from the study and were debriefed after participation. No identifying information was available about participants during or after data collection.

Sample characteristics are reported in Table 1. Of the participating mothers ($N$ = 176), data were excluded due to incomplete responses, i.e., the participant did not complete the survey and were therefore not included ($N$ = 10), gestation age <37 weeks ($N$ = 9), child developmental disorder ($N$ = 2), child outside the study age range ($N$ = 2) and non-birth mother ($N$ = 1). One response was excluded from data analysis due to outliers 3 SD below the mean on the Parent Infant Caregiving Touch Scale (PICT) [70]. This resulted in a final sample of 151 women with a 6- to 18-month-old child ($M_{childage}$ = 11.4, $SD$ = 3.57, months, female = 81, male = 70).

*Parent-Infant Caregiving Touch Scale (PICT; Koukounari et al., 2015).* Our primary outcome variable of interest was the frequency caregiving behaviours reported on the PICT. Participants were asked "How often do you find yourself doing each of the following things with your baby?". Responses consisted of a 5-point Likert scale with levels coded from 1 (Never) to 5 (A Lot). In the original validation, the 12 items followed a clear two-factor structure: Holding (4 items: "*I hold/pick up/cuddle/rock my baby*") and Stroking (4 items: "*I stroke my baby's face/back/tummy/arms and legs*). A third factor called Affective communication (4 items: "*I kiss/talk to/watch my baby/I leave my baby to lie down*") was less robust. A previous study found that parental reports of stroking on the PICT moderately converged with diary reports of

**Table 1. Study 1 sample descriptive statistics.**

|  | N = 151 |
| --- | --- |
| **Child age (months)** |  |
| Mean (SD) | 11.4 (3.57) |
| Median [Min, Max] | 12.0 [6.00, 18.0] |
| **Child gender** |  |
| Male | 69 (45.7%) |
| Female | 81 (53.6%) |
| Prefer not to say | 1 (0.7%) |
| **Child ethnicity** |  |
| Asian/Asian British Irish | 3 (2.0%) |
| Black/African/Caribbean/Black British Irish | 0 (0.0%) |
| Mixed ethnicity: White and other | 15 (9.9%) |
| Other | 9 (6.0%) |
| Prefer not to say | 2 (1.3%) |
| White British/Irish | 122 (80.8%) |
| **Highest education** |  |
| GCSE or similar | 11 (7.3%) |
| A level or similar | 14 (9.3%) |
| University degree or similar | 91 (60.3%) |
| Postgraduate degree or similar | 29 (19.2%) |
| Prefer not to say | 6 (4.0%) |
| **Household income** |  |
| £0-£14,000 | 1 (0.7%) |
| £14,001-£24,000 | 5 (3.3%) |
| £24,001-£42,000 | 28 (18.5%) |
| £42,001 or more | 103 (68.2%) |
| Prefer not to say | 14 (9.3%) |
| **Other children** |  |
| Multiparous (1 or more children) | 92 (60%) |
| Primiparous | 59 (40%) |

stroking over one week [76]. Since we were interested in the types of caregiving behaviours that may be involved in maternal interoception, and their frequency, a mean score was calculated for each subscale (min = 4, max = 20 for each subscale; Holding, Stroking and Affective Communication).

*Multidimensional Assessment of Interoceptive Awareness Version 2 (MAIA-2).* The MAIA-2 was selected as a measure of self-reported interoceptive focus because it taps into distinct dimensions which reflect participants' tendency to focus on bodily changes and adapt to these changes using regulatory strategies. Other self-report measures, such as the Body Perception Questionnaire [71], measure awareness of specific interoceptive sensations, i.e., stomach tension. The MAIA-2 allowed us to explore, more broadly, the potential contributions of different interoceptive dimensions that may be involved in caregiving and in linking maternal emotion regulation to caregiving [30].

The MAIA-2 contains 37 items coded from 0 (Never) to 5 (Always) and follows an eight-factor structure (see Table 2 for details). Greater *Attention-Regulation*, *Self-Regulation* and *Body Listening* reflect adaptive styles of attention to, and regulation of, interoceptive states, whereas *Not-Worrying* and *Not-Distracting* may reflect clinically significant styles of anxious

**Table 2. Factor structure and example item MAIA-2.**

| Factor description | Example item |
|---|---|
| *Noticing (4 items, min = 0, max = 20)*: Awareness of uncomfortable, comfortable, and neutral body sensations | "I notice changes in my breathing, such as whether it slows down or speeds up" |
| *Listening (3 items, min = 0, max = 15)*: Active listening to the body for insight | "I listen to my body to inform me about what to do" |
| *Emotional Awareness (5 items, min = 0, max = 25)*: Awareness of the connection between body sensations and emotional states | "I notice how my body changes when I am angry" |
| *Attention-Regulation (7 items, min = 0, max = 35)*: Ability to sustain and control attention to body sensations | "I can refocus my attention from thinking to sensing my body" |
| *Self-Regulation (4 items, min = 0, max = 20)*: Ability to regulate distress by attention to body sensations | "I can use my breath to reduce tension" |
| *Not-Distracting (6 items, min = 0, max = 30)*: Tendency not to ignore or distract oneself from sensations of pain or discomfort | "When I feel pain or discomfort, I try to power through it" |
| *Not-Worrying (5 items, min = 0, max = 25)*: Tendency not to worry or experience emotional distress with sensations of pain or discomfort | "I start to worry that something is wrong if I feel any discomfort" |
| *Trusting (3 items, min = 0, max = 15)*: Experience of one's body as safe and trustworthy | "I trust my body sensations" |

or avoidant behaviour [72]. A score for each subscale is calculated by averaging the scores of its individual items (see Table 2 for details).

*Body Understanding Measure for Pregnancy (BUMPS).* BUMPS [67] is a 19-item scale capturing mothers' feelings toward their body that are unique to pregnancy. The original scale was administered to women who were at different stages of pregnancy at the time (i.e., first, second and third trimesters). In our study, we asked mothers to form a retrospective answer based on how they felt when they were pregnant with their baby (6 to 18 months old at the time of participation) not in relation to a specific trimester or period in time. Therefore, we reworded the items to correspond to the past tense (e.g., "I felt good about my changing body"). Responses are coded from 1 (Never) to 5 (A Lot), and a total BUMPS score is calculated by summing all items (min = 19, max = 95). Positively worded items were reversed; a higher score is indicative of greater body *dissatisfaction* during pregnancy.

*Social Touch Questionnaire (STQ).* This 20-item scale [73] measures participants' attitudes towards tactile behaviours across various everyday situations, such as scenarios involving touch with familiar or unfamiliar people (e.g., "I feel uncomfortable when someone I don't know very well hugs me") on a 5-point Likert scale with levels coded from 0 (Not at all) to 4 (Extremely). Several items are reverse coded, meaning a higher score reflects greater aversion to social touch.

*UK-Communicative Development Inventory Words and Gestures (UK-CDI).* The UKI-CDI [69] is an inventory of words that infants, aged 8 to 18 months old, may understand and produce. The body-part section contains 23 body-part nouns, enabling us to obtain measures of receptive and expressive vocabulary for words relating to body parts in the 8- to 18-month-olds in our sample ($N$ = 124). Receptive vocabulary measures the total number of words that a child understands. Expressive vocabulary measures the total number of words that a child understands and says.

**Analytical plan.** All analyses were performed in R Studio (v. 4.0.2). We examined Cronbach's alpha to assess the reliability of the measured scales, and where $\alpha < .60$, we assessed the underlying structure using Confirmatory Factor Analysis. We examined the distribution of each measure for the following correlation and regression analyses, conducted to assess relationships between our outcome variables of interest (caregiving behaviours on the PICT) and

maternal self-reported interoceptive focus (MAIA-2 subscales), pregnancy body satisfaction (BUMPS) and attitudes to social touch (STQ). Initially we had intended to classify infants into three age groups but based on previous reviewer recommendations we treated child as a continuous variable. We checked the distribution of vocabulary comprehension before conducting a logistic mixed-effects model to estimate infants' body-part word comprehension from engagement in caregiving behaviours, controlling for child age as a continuous variable.

## Results

**Scale reliability.** *PICT, MAIA-2, BUMPS, STQ.* The internal consistency for the MAIA-2 and its subscales ranged between $\alpha = .62$ and $.90$, consistent with previous studies [74, 75]. The BUMPS and STQ showed excellent reliability ($\alpha = .91$). Across the three PICT factors, Stroking and Holding had good reliability ($\alpha = .82$, $\alpha = .79$, respectively) but Affective Communication had poor reliability ($\alpha = .36$), as found in the original validation [70].

Confirmatory factor analysis was conducted to examine the underlying structure of the PICT's latent variables (see S1 File), using the *lavaan* package (v0.6–11). Inspection of the estimates showed poor loadings ($< .40$) for most items on the latent variable measuring Affective Communication (between .13 and .34), and on the Holding factor, *I rock my baby* had a loading of .18.

Moving forward, we dropped the Affective Communication factor from our analyses as it was unreliable. Our modified Holding factor no longer contained *I rock my baby* (i.e., *I hold/pick up/cuddle my baby*). Estimates showed that *I rock my baby* cross-loaded onto both Affective Communication and Stroking, thus, we continued with the original Stroking factor (i.e., *I stroke my baby's face/back/tummy/arms and legs*) but carried out exploratory analyses with a modified Stroking/Rocking factor (containing *I rock my baby*) (see S2 File), whose reliability remained equivocal to the original Stroking factor ($\alpha = .81$). The reliability for the modified Holding factor slightly improved ($\alpha = .83$). A mean score was calculated for each modified subscale (Holding; min = 3, max = 15, Stroking/Rocking; min = 5, max = 25).

**Descriptive statistics.** *PICT, MAIA, BUMPS, STQ.* Descriptive statistics for all questionnaires included can be found in Table 3. Visualisation of the PICT subscales suggested that scores for Stroking were normally distributed but mothers reported disproportionately high frequencies of Holding behaviours. Across the items, the majority of participants answered *Always* or *Most of the time* for holding (98.6%), picking up (96%) and cuddling (99.3%). None

**Table 3. Study 1: Descriptive statistics for all questionnaire measures.**

| Questionnaire (number items) | Mean (SD) | Min | Max | Possible range |
|---|---|---|---|---|
| PICT–Holding (3) | 14.2 (1.21) | 10 | 15 | 3–15 |
| PICT–Stroking (4) | 15.7 (2.94) | 8 | 20 | 4–20 |
| PICT–Stroking/Rocking (5) | 19.1 (3.57) | 8 | 25 | 4–25 |
| MAIA-2 –Noticing (4) | 3.14 (0.76) | 1.50 | 4.75 | 0–9 |
| MAIA-2 –Listening (3) | 2.12 (0.95) | 1 | 5 | 0–8 |
| MAIA-2 –Emotional Awareness (5) | 3.13 (0.91) | 1 | 6 | 0–10 |
| MAIA-2 –Attention Regulation (7) | 2.52 (0.81) | 1.29 | 6 | 0–12 |
| MAIA-2 –Self-Regulation (4) | 2.57 (0.93) | 1 | 5 | 0–9 |
| MAIA-2 –Not Distracting (6) | 2.88 (0.81) | 1 | 4.67 | 0–11 |
| MAIA-2 –Not Worrying (5) | 3.36 (0.69) | 1.20 | 5.80 | 0–10 |
| MAIA-2 –Body Trusting (3) | 3.11 (1.09) | 1 | 5 | 0–8 |
| BUMPS (20) | 55 (16) | 19 | 95 | 19–95 |
| STQ (20) | 52.5 (13.5) | 25 | 87 | 0–100 |

reported *Never* or *Rarely* and only five participants responded with *Sometimes*. Given the lack of variance in Holding, we did not include this factor in our analyses moving forward. All scores were standardised using z-scores for the following analysis across different measurement scales. There was no relationship between child age and mother-infant stroking ($r = 0.068$, $p = .405$), consistent with a previous study using PICT with 6- to 13-month-olds [76].

*Comprehension of body-part vocabulary (UK-CDI).* The total number of words was not normally distributed. For 8-month-olds ($N = 18$), most words (94%) were recorded as not understood ($M = 0.056$, $SD = 0.24$, min = 0, max = 1) and for 9-month-olds ($N = 9$), 80% of the words were recorded as not understood ($M = 1$, $SD = 2.00$, min = 0, max = 5). By 10 months old ($N = 9$), the number of words recorded as not understood decreased to 66%. Published norms for the UK-CDI are available only for the full inventory of words, however, comparing against the American-English equivalent [77] which includes 20 of the same body-part words, the proportion of 8-month-olds in our sample who showed some comprehension of body-part words was lower (6% vs. 10%). To avoid floor effects, we did not include 8- and 9-month-olds in the analysis as these were highly skewed. Of the total 23 words, 10- to 18-month olds' comprehension was highest for hand (47.40%), tummy (46.40%), feet (42.3%) and nose (41.2%). See S3 File for the distribution of responses for all body part words.

**Associations between maternal self-reported interoception and mother-infant stroking, pregnancy body satisfaction and social touch attitudes.** Bivariate correlations can be found in Table 4. Concerning our first key hypothesis, weak positive correlations suggested that mothers who reported engaging in higher frequencies of stroking also reported a higher scores on the subscales *Noticing* ($r = .19$, p = .020), *Listening* ($r = .17$, p = .033) and *Self-Regulation* ($r$

**Table 4. Bivariate correlations between mother-infant stroking and MAIA-2, BUMPS and STQ.**

| | Stroking PICT | Noticing MAIA | Body Listen MAIA | Self-Reg MAIA | Attention Reg MAIA | Body Trust MAIA | Emotion Aw MAIA | Not Distract MAIA | Not worry MAIA | BUMPS | STQ |
|---|---|---|---|---|---|---|---|---|---|---|---|
| *Stroking PICT* | | | | | | | | | | | |
| *Noticing MAIA* | **0.190** * **(.020)** | | | | | | | | | | |
| *Body Listen MAIA* | **0.173** * **(.033)** | 0.408*** (< .001) | | | | | | | | | |
| *Self-reg MAIA* | **0.187** * **(.022)** | 0.301*** (< .001) | 0.634*** (< .001) | | | | | | | | |
| *Attention Reg MAIA* | 0.076 (.351) | 0.388*** (< .001) | 0.593*** (< .001) | 0.614*** (< .001) | | | | | | | |
| *Body Trust (MAIA)* | 0.037 (.653) | 0.309*** (< .001) | 0.500*** (< .001) | 0.543*** (< .001) | 0.480*** (< .001) | | | | | | |
| *Emotion Aw MAIA* | 0.130 (.111) | 0.338*** (< .001) | 0.611*** (< .001) | 0.500*** (< .001) | 0.477*** (< .001) | 0.385*** (< .001) | | | | | |
| *Not Distract MAIA* | -0.045 (.582) | 0.262*** (.001) | 0.227*** (.005) | 0.170*** (.037) | 0.199*** (.014) | 0.361*** (< .001) | 0.125 (.125) | | | | |
| *Not Worry MAIA* | -0.021 (.794) | 0.073 (.375) | 0.064 (.434) | 0.211** (.009) | 0.185* (.023) | 0.238** (.003) | -0.029 (.723) | -0.063 (.442) | | | |
| *BUMPS* | -0.009 (.911) | -0.119 (.145) | **-0.279*** (.001)** | **-0.255*** (.002)** | **-0.205*** (.012)** | **-0.406*** (< .001)** | **-0.251*** (.002)** | **-0.215*** (.008)** | -0.050 (.546) | | |
| *STQ* | -0.091 (.266) | **-0.164* (.044)** | -0.138 (.091) | **-0.166* (.042)** | -0.090 (.274) | **-0.307*** (< .001)** | **-0.171* (.035)** | **-0.316*** (< .001)** | -0.047 (.565) | **0.163* (.045)** | |

Computed correlation used pearson-method. Bold fonts indicate statistically significant variables of interest (p-value in brackets).

= .19, p = .022). These correlations remained when using our modified Stroking/Rocking factor (see S2 File). The other subscales of the MAIA were not related to engagement in stroking ($rs$ = -.009 to .13, $ps$ = .911 to .111) (see Table 4). There was no significant relationship between mothers' engagement in stroking and their (retrospective) body satisfaction during the pregnancy as reported in BUMPS ($rho$ = -.009, $p$ = .911).

Our second hypothesis was that body dissatisfaction during pregnancy (BUMPS) would be negatively correlated with maternal interoceptive focus. Correlations suggested that mothers who were less satisfied with their body during pregnancy reported significantly poorer body *Listening*, *Trusting*, *Attention-Regulation*, *Self-Regulation* and *Distraction from bodily states* ($rs$ = -.215 to -.406, $ps$ = .012 to < .001), consistent with Kirk and Preston's [67] original finding.

Mothers who showed negative social touch attitudes reported poorer *Noticing*, *Self-Regulation*, *Emotional Awareness*, *Trusting* and *Distraction from bodily states* ($r_s$ = -.164 to -.316, $ps$ = 0.046 to < .001). We found no evidence for a relationship between social touch attitudes and engagement in stroking ($r$ = -.09, $p$ = 0.266).

**Associations between caregiving and infants' body part comprehension.** In our sample of 10- to 18-month-olds, there was a positive association between infants' body-part comprehension and mother-infant stroking ($r_s$ = .27, $p$ = .007) and as expected, there was a positive correlation between child age and body part comprehension ($r_s$ = .54, $p$ = < .001). A positive correlation was found between mother-infant stroking and child age in the 10- to 18-month-olds ($r_s$ = .22, p = .031). The relationship between stroking and infants' comprehension of body-part words remained after controlling for child age *($r_s$ = .20, $p$ = .041)*.

**Predicting retrospective body dissatisfaction during pregnancy (BUMPS) from self-reported interoception (MAIA-2).** A model containing *Listening*, *Attention Regulation*, *Self-Regulation*, *Not Distracting and Body Trusting* was built to explain variation in retrospective body satisfaction across pregnancy. All predictors were entered simultaneously. The model explained 17% of the variance, $F(5, 145)$ = 6.30, $p$ = < .001, and Body Trusting emerged as the only significant predictor, with increased body trust predicting lower body dissatisfaction during pregnancy (see Table 5).

**Predicting social touch attitudes (STQ) from self-reported interoception (MAIA-2).** A model was built to predict social touch attitudes from *Noticing*, *Self-Regulation*, *Emotional Awareness*, *Not Distracting* and *Body Trusting*. The model explained a statistically significant and moderate proportion of variance ($R^2$ = 0.15, $F(5, 145)$ = 5.02, $p$ < .001). *Not Distracting* and *Body Trusting* emerged as significant predictors (see Table 6).

**Predicting body-part vocabulary comprehension in 10- to 18-month-olds from mother-infant stroking.** A logistic mixed effects model tested whether 10- to 18-month-olds' body-

**Table 5. Summary of multiple linear regression predicting retrospective body satisfaction during pregnancy.**

| Predictors | Effects of interoceptive focus on retrospective body dissatisfaction during pregnancy | | |
|---|---|---|---|
| | *Estimates* | *CI* | *p* |
| (Intercept) | -0.00 | -0.15 – 0.15 | 1.000 |
| BodyListening | -0.11 | -0.32 – 0.10 | 0.298 |
| AttentionRegulation | 0.05 | -0.15 – 0.25 | 0.637 |
| SelfRegulation | -0.02 | -0.24 – 0.20 | 0.860 |
| NotDistracting | -0.07 | -0.23 – 0.09 | 0.361 |
| BodyTrusting | -0.34 | -0.53 – -0.14 | **0.001** |

Observations 151

$R^2$ / $R^2$ adjusted 0.178 / 0.150

**Table 6. Summary of multiple linear regression predicting social touch attitudes.**

| Predictors | Effects of interoceptive focus on social touch attitudes | | |
| --- | --- | --- | --- |
| | Estimates | CI | p |
| (Intercept) | -0.00 | -0.15 – 0.15 | 1.000 |
| Noticing | -0.02 | -0.19 – 0.14 | 0.772 |
| Self-Regulation | 0.03 | -0.17 – 0.22 | 0.792 |
| Emotional Awareness | -0.07 | -0.25 – 0.11 | 0.456 |
| Not Distracting | -0.23 | -0.40 – -0.07 | **0.006** |
| Body Trusting | -0.20 | -0.40 – -0.01 | **0.040** |

Observations 151

$R^2$ / $R^2$ adjusted 0.148 / 0.118

part vocabulary comprehension could be predicted by maternal engagement in stroking. We used the *lme4* package v.1.1–26 [78] following guidance from [79]. The model predicted infants' body part comprehension (understands/does not understand) of all 23 words as a function of stroking and child age in months, while modelling participant ID ($N = 97$) as a random effect. A likelihood ratio test indicated that the model including the fixed predictors provided a better fit for the data than a model without them ($\chi^2 = 47.26$; $p < .001$, $N = 97$). As expected, older children had significantly greater knowledge of body part words ($B = .88$, $SE = 0.14$, $t = 6.22$, $p < .001$). The frequency of stroking was a significant predictor of 10- to 18-month-old infants' comprehension of body part words ($B = .54$, $SE = .28$, $t = 2.0$, $p = .049$).

## Discussion of Study 1

The key findings of Study 1 was that mothers who reported greater tendencies to focus on, and regulate, their own interoceptive states reported engaging in more stroking and rocking. Conversely, engagement in these caregiving behaviours was not associated with pregnancy body satisfaction, nor social touch attitudes. Aversion to social touch correlated with unique aspects of interoception, i.e., negative styles of distraction and trust towards bodily states. Importantly, these were not relevant to stroking, potentially explaining the lack of relationship. We discuss the interpretation of these results in greater detail in the General Discussion.

We designed and implemented Study 2 to extend the potential role of interoception by investigating whether objective interoceptive accuracy also plays a role in maternal engagement in stroking and rocking. We also aimed to replicate our previous finding that mothers who show greater self-reported interoceptive focus also report greater engagement in mother-infant stroking and rocking.

## Study 2

### Materials and methods

**Participants.** We aimed for the same sample size as outlined in Study 1, i.e., 146, however, due to stricter inclusion criteria (outlined below) it was challenging to recruit the same number of participants. We managed to recruit 117 participants and the final sample were 111 women with a 6- to 18-month-old child ($M_{\text{child age}} = 10.8$ months, $SD = 4.58$; 62 female, 49 male). Sample characteristics are reported in Table 7. Using Prolific (https://prolific.co/) in April and May 2021, we screened female participants between 18 and 40 years old, with a child born in the year 2020, who were residing in the United Kingdom, Ireland or the United States at the time. As the heartbeat counting task was conducted online (details below), participants were

**Table 7. Study 2: Sample descriptive statistics.**

| | N = 111 |
|---|---|
| **Child age (months)** | |
| Mean (SD) | 10.8 (4.58) |
| Median [Min, Max] | 11.0 [6, 18.0] |
| **Child gender** | |
| Male | 49 (44.1%) |
| Female | 62 (55.9%) |
| **Mother age (years)** | |
| Mean (SD) | 31.5 (3.71) |
| Median [Min, Max] | 31.0 [22.0, 40.0] |
| **Child ethnicity** | |
| Asian/Asian British | 3 (2.7%) |
| Black/African/Caribbean/Black British | 7 (6.3%) |
| Mixed ethnicity: White and other | 18 (16.2%) |
| Other (please specify) | 3 (2.7%) |
| Prefer not to say | 1 (0.9%) |
| White British/Irish | 79 (71.1%) |
| **Highest education** | |
| GCSE or similar | 7 (6.3%) |
| A level or similar | 19 (17.1%) |
| University degree | 47 (42.3%) |
| Postgraduate degree | 38 (34.2%) |
| Prefer not to say | 0 (0%) |
| **Household income** | |
| £0 - £14,000 | 6 (5.4%) |
| £14,001 - £24,000 | 9 (8.1%) |
| £24,001 - £42,000 | 26 (23.4%) |
| £42,001 or more | 65 (58.6%) |
| Prefer not to say | 5 (4.5%) |
| **Other children** | |
| Multiparous (1 or more children) | 62 (53%) |
| Primiparous | 55 (47%) |

screened for access to a computer or laptop, and consent to be video recorded. Of 380 eligible participants in Prolific, 113 agreed to take part and a further 4 participants were recruited via social media ($N$ = 117). Several were excluded due to incomplete responses ($N$ = 2), gestation age < 37 weeks ($N$ = 2), child developmental disorder ($N$ = 2). We identified no outliers when using a criterion of 3 +/- SD above the mean, resulting in a final sample of 111 participants. The study was hosted on Gorilla Experiment Builder (http://www.gorilla.sc/) and was approved by the Institutional Ethics Committee at Royal Holloway University of London [ref: 2621]. On giving informed written consent, participants were free to withdraw from the study at any time and were debriefed after participation. No identifying information was available about participants during or after data collection.

**Questionnaires.** Participants completed all subscales of the PICT [70] to establish whether the factor structure was unreliable, as found in Study 1. Participants completed the MAIA-2 subscales of interest which previously showed positive relationships to mothers' engagement in stroking in Study 1, i.e., *Noticing*, *Listening* and *Self-Regulation* MAIA-2 [14] as well as the

*Attention-Regulation* subscale, because it previously showed associations with greater interoceptive accuracy in the heartbeat counting task [80, 81]. In addition, given the interesting findings in Study 1 regarding pregnancy body satisfaction, we were interested in exploring postnatal and current body satisfaction. There is currently no validated scale for measuring postnatal body satisfaction, thus participants completed the Body Image States Scale (BISS) [82] as a retrospective self-report to capture body satisfaction during the final trimester of pregnancy (12 weeks before birth), the first 12 weeks postpartum, and at present. Participants responded to six items concerning their satisfaction towards their physical appearance, body size and shape, weight and attractiveness, on a 9-point scale. For retrospective accounts, participants were asked to reflect on the experience of their body during those periods. A total score was computed for each time point, by summing the six items.

**Online heartbeat counting task: Remote photoplethysmography.** Participants completed a measure of interoceptive accuracy that could be obtained remotely. Specifically, the Heartbeat Counting Task (HCT) [13] was integrated with a novel method for calculating heart rate from pixel changes in video recordings of human skin [83]. Remote photoplethysmography (rPPG) uses a light source to measure the variation in oxygenated blood flow and corresponds to phasic changes in heart rate [84]. rPPG works via photo-amplification, which detects variations in the reflected colours of the skin caused by changes in capillary tissue movement that can be extracted from luminance values in the video recording [85]. An algorithm applies signal detection procedures based a step-by-step approach [83], most recently used in [86, 87]. First, the algorithm automatically detects the presence of a face and extracts facial features, filtering out other irrelevant ones, i.e. hair, teeth, clothing, background. For every video frame and RGB colour channel, the average of all selected pixels was computed and aggregated across time. A filter (0.04Hz) is applied to the raw signal to remove movement artefacts and low-frequency changes in the signal. The most prominent component signal in all colour channels is extracted and converted to a power spectra estimation using fast-Fourier transformation. A high peak in power at a given frequency reflects the heart rate. Low-pass frequency filter was applied to the power spectra (0.02Hz) and the heart rate at the highest power peak across components was selected as the final rPPG heart rate. In a validation study using the rPPG method presented in the current study, Di Lernia [87] showed that heart rate estimates from webcam video recordings converged well with heart rate estimates extracted from a validated mobile application.

Adopting the rPPG allowed a measurement of heart rate that is required for the measure of interoceptive (cardiac) accuracy in the HCT. Specifically, during the HCT, participants are instructed to focus and count their heartbeats during several short intervals (e.g. 25 s, 35 s, 45 s), without taking their pulse and without guessing. To quantify interoceptive accuracy in the HCT, participants' reported heartbeats are compared against their actual heartbeats (in this case extracted via the rPPG).

In experimental trials, the webcam was activated and participants were instructed to silently count their heartbeats without taking their pulse and were instructed not to guess. A tone indicated when they should start counting, and a second when they should stop (webcam deactivated). Participants then reported the number of heartbeats they felt and rated how confident they felt about their response, using a sliding scale between 0 and 100. Participants completed the task in three intervals (25s, 35s, 45s) and the trial order was randomised. Heart rate estimation was conducted offline using the video recordings (using the algorithmic approach outlined above). Because HR estimation requires video recordings from a webcam and enough luminance values, participants were instructed to sit still in a room with natural light covering their face. Before the task, participants had to successfully answer questions regarding requirements to proceed to the experiment, e.g., "I can complete the study on a mobile phone"

(correct answer: no), "I can perform the experiment at night in a dark room" (correct answer: no) and so forth. The webcam was activated between each experimental trial so that participants could adjust themselves accordingly and they were given one practice trial at the beginning.

In each trial, reported and actual heartbeats were compared to calculate an index of interoceptive accuracy (Iacc) using the following equation: (1 –(|Actual number of heartbeats–number of heartbeats reported|/Actual number of heartbeats)). An index of mean Iacc was calculated for each participant by averaging across the three trials. Finally, participants were asked to estimate their resting heart rate, to control for knowledge of heart rate.

**Analysis plan.**    All analyses were performed in R Studio (v. 4.0.2). The analysis followed the same protocol as Study 1, beginning by examining Cronbach's alpha to assess the reliability of the measured scales. Where $\alpha < .60$, we assessed the underlying structure using Confirmatory Factor Analysis (see S4 File). We examined the distribution of each measure for the following correlation and regression analyses, conducted to assess relationships between our outcome variables of interest (caregiving behaviours on the PICT) and maternal self-reported interoceptive focus (MAIA-2 subscales) and postnatal body satisfaction (BISS). For analysis of interoceptive accuracy, the video recordings from the heartbeat counting task were visually inspected for eligibility in the algorithm's analysis in estimating heart rate. Scores for interoceptive accuracy were calculated and the distribution examined. We checked reliability of the algorithm across trials, before conducting correlations to assess the relationship between interoceptive accuracy and caregiving (stroking and rocking).

*Data exclusion*. One participant was excluded for taking her pulse during the experiment as seen on the video recording. Others produced video recordings which could not be processed by the algorithm, because of poor lighting conditions ($N = 7$), sitting too far away ($N = 3$), light reflecting on glasses ($N = 3$) or a corrupt video file ($N = 1$). Forty-five participants were later excluded because one or more video recordings did not meet the eligibility criteria for a reliable heart rate detection [83], including frame rate below 20 ($N = 5$), heart rate outside 50 to 120 BPM ($N = 2$), a difference of 10 BPM or higher between consecutive trials ($N = 8$) and unreliable peak detection ($N = 30$). The remaining sample for the analysis involving cardiac perception consisted of 52 participants. For the correlation between interoceptive accuracy and caregiving, three participants were excluded due to child gestation age $< 37$ weeks ($N = 2$) and child developmental disorder ($N = 1$), leaving a final sample of 49 participants in this analysis.

## Results

**Reliability of variables.**    *PICT*, *MAIA-2*, *BISS*. Internal consistency for the MAIA-2 subscales (*Noticing*, *Listening*, *Attention-Regulation*, *Self-Regulation*) ranged between $\alpha = .68$ and .85, similar to levels we found previously. The BISS showed excellent reliability ($\alpha = .95$) for each time point. Cronbach's alpha identified similar levels of reliability for the PICT subscales as in Study 1. Stroking had good reliability ($\alpha = .83$; Study 1 $\alpha = .82$). The reliability was also comparable for Holding ($\alpha = .77$; Study 1 $\alpha = .79$). Affective communication had higher reliability ($\alpha = .49$; Study 1 $\alpha = .36$) but was still poor.

Confirmatory factor analysis assessed the underlying structure of the latent variables and suggested the model did not fit the data well (see S4 File). Inspection of the estimates revealed several poor loadings for items on Affective Communication (*I leave my baby to lie down* = .25, *I kiss my baby* = .18), although, *I watch my baby* and *I talk to my baby* had high loadings (.91 and .82, respectively), unlike in Study 1, suggesting the Affective Communication factor is unreliable. As in Study 1, *I rock my baby* loaded more strongly onto the Stroking factor. We

**Table 8. Study 2: Descriptive statistics for all questionnaire measures (N = 111).**

| Questionnaire (number items) | Mean (SD) | Min | Max | Possible range |
|---|---|---|---|---|
| PICT–Holding (3) | 14.2 (1.21) | 10 | 15 | 3–15 |
| PICT–Stroking (4) | 15.7 (2.94) | 8 | 20 | 4–20 |
| MAIA-2 –Noticing (4) | 3.47 (0.77) | 1.25 | 5 | 0–9 |
| MAIA-2 –Listening (3) | 4.17 (1.39) | 1.50 | 7.50 | 0–8 |
| MAIA-2 –Self-regulation (4) | 4.44 (1.43) | 1.60 | 7.60 | 0–9 |
| MAIA-2 –Attention Regulation (7) | 4.71 (1.33) | 2.25 | 7.50 | 0–12 |
| BISS–Final trimester (6) | 28.2 (11.80) | 6 | 50 | 6–54 |
| BISS– 12 weeks postpartum (6) | 21.6 (10.20) | 6 | 45 | 6–54 |
| BISS–Current [last 2 weeks] (6) | 24.6 (12.00) | 6 | 51 | 6–54 |

*rPPG: Interoceptive accuracy (N = 52).* The interoceptive accuracy data, with a possible score between 0 and 1, were negatively skewed in our sample. Interoceptive accuracy (*M* = 0.67, *SD* = 0.22, median = .73) is in line with previous studies of interoceptive accuracy in women [88]. Participants' estimate of their resting heart rate did not correlate with interoceptive accuracy ($r_s$ = -.03, *p* = .83), suggesting that people's beliefs about their heart rate did not influence interoceptive accuracy. Average heart rate was 71.2 beats per minute (*SD* = 8.36, median = 70.6, min = 54.5, max = 93.4) (see S5 File).

dropped the Affective Communication factor from our analyses, and removed *I rock my baby* from the Holding factor, which slightly improved its reliability (α = .78).

*rPGG: Interoceptive accuracy.* To test the reliability of the rPPG algorithm in estimating heart rate, we checked the variability in HR and Iacc across the experimental trials (25s, 35s, 45s). There were no significant differences in HR ($F(2,153)$ = 0.007, *p* = 0.99) and evaluation of the residual plots suggested they met assumptions for linearity, normality and homoskedasticity. For the Iacc data, several outliers (i.e., two participants reporting 0 heartbeats) meant the assumption of normality was violated and a Kruksal-Wallis test was used, which identified no significant differences in Iacc across trials ($\chi^2$ = 0.03, *df* = 2, *p* = 0.983), suggesting good reliability of the algorithm.

**Descriptive statistics.** *PICT, MAIA-2, BISS.* Descriptive statistics for all included questionnaires can be found in Table 8. Consistent with Study 1, Stroking was normally distributed, but mothers' engagement in Holding was negatively skewed. Mothers tended to report high frequencies of holding behaviours and thus we decided not to include it in our analyses due to the lack of variance, as in Study 1. The MAIA-2 subscales (*Noticing, Listening, Self-Regulation* and *Attention-Regulation)* were normally distributed. Unlike BUMPS in Study 1, where higher scores present greater body *dissatisfaction*, higher scores on the BISS indicates greater body satisfaction.

**Associations between maternal self-reported interoception, mother-infant stroking, pregnancy body satisfaction.** Bivariate correlations for the MAIA-2 subscales, mother-infant stroking and retrospective BISS scores in the full sample (*N* = 111) can be found in Table 9. First, we replicated the key findings of Study 1 that mothers who reported engaging in more stroking reported greater body *Listening* (*r* = .19, *p* = .044) and *Self-Regulation* (*r* = .18, *p* = .048). The relationship between mother-infant stroking and *Noticing* was in the same direction but did not reach statistical significance (*r* = .17, *p* = .074). When analysing relationships between Stroking/Rocking and MAIA-2 subscales, there was a positive association with *Noticing* (*r* = .20, *p* = .032).

Contrary to Study 1, we did not find associations between self-reported interoception on the MAIA-2 subscales and mothers' retrospective body satisfaction when referring to the final trimester (*rho*s = .06 to .15, *p*s = .208 to .520) or the first 12 weeks postpartum (*rho*s = -.04 to *p*s = .118 to .712). However, the subscales *Listening, Self-Regulation, Attention-Regulation* were

**Table 9. Bivariate correlations between stroking (PICT), maternal self-reported interoceptive focus (MAIA-2 subscales) and body satisfaction (BISS) in full sample (N = 111).**

| | Stroking PICT | Noticing MAIA-2 | Body Listening MAIA-2 | Self-Reg MAIA-2 | Attention-Reg MAIA-2 | Final Trimester BISS | Postnatal BISS | Current BISS |
|---|---|---|---|---|---|---|---|---|
| *Stroking PICT* | | | | | | | | |
| *Noticing MAIA-2* | 0.170 *(.074)* | | | | | | | |
| *Body Listening MAIA-2* | **0.191* (.045)** | 0.595*** *(< .001)* | | | | | | |
| *Self-Reg MAIA-2* | **0.184* (.048)** | 0.498*** *(< .001)* | 0.580*** *(< .001)* | | | | | |
| *Attention-Reg MAIA-2* | 0.098 *(.308)* | 0.589*** *(< .001)* | 0.720*** *(< .001)* | 0.647*** *(< .001)* | | | | |
| *Final Trimester BISS* | -0.090 *(.347)* | 0.062 *(.520)* | 0.122 *(.203)* | 0.121 *(.208)* | 0.088 *(.358)* | | | |
| *Postnatal BISS* | 0.032 *(.741)* | -0.035 *(.712)* | 0.113 *(.236)* | 0.149 *(.118)* | 0.082 *(.392)* | 0.583*** *(< .001)* | | |
| *Current BISS* | 0.002 *(.982)* | 0.124 *(.195)* | **0.397*** (< .001)** | **0.333*** (< .001)** | **0.302*** (.001)** | 0.205* *(.031)* | 0.287** *(.002)* | |

Computed correlation used pearson-method. Bold fonts indicate statistically significant variables of interest (p-value in brackets).

moderately negatively correlated with body satisfaction at present (*rho*s = -.30 to -.40, *p*s = < .001) while *Noticing* showed no association (*rho* = 0.12, *p* = .195).

When analysing relationships between stroking and interoceptive accuracy in the subsample for whom we had valid data (*N* = 49), the results showed that maternal interoceptive accuracy did not correlate with engagement in stroking (*r* = -.09, *p* = .526). For the same subsample (*N* = 49), when examining the relationships between stroking and self-reported interoception on the MAIA-2 subscales, results showed that the relationship between mother-infant stroking and *Listening* was significant (*r* = .31, *p* = .030), but not statistically significant for *Noticing* (*r* = .22, *p* = .13) or *Self-Regulation* (*r* = .25, *p* = .070), likely due to a lack of power (*N* = 49). Interoceptive accuracy correlated with *Attention-Regulation* on the MAIA-2 (*r* = .34, *p* = .01, *N* = 52), as found previously in laboratory-based studies using the heartbeat counting task in person [80].

**Predicting mother-infant stroking from self-reported interoceptive focus across Study 1 and Study 2 (N = 262).** To identify the best predictors of maternal engagement in stroking from *Noticing*, *Listening* and *Self-Regulation* across the combined samples of Study 1 and Study 2 (*N* = 262) that give us increased power, multivariate linear regression was conducted using a stepwise method. A model containing *Self-Regulation* and *Noticing* provided the best fit, explaining a statistically significant proportion of variance (5%) of maternal engagement in stroking, *F*(2, 259) = 6.62, *p* = .002. *Body Listening* did not significantly increase the variance so was not included in the final model. *Self-Regulation* and *Noticing* were significant predictors of mother-infant stroking (see Table 10). A model containing *Noticing* and *Self-Regulation* to estimate variance in Stroking/Rocking across the combined sample was also significant (see S2 File).

**Retrospective analysis of body dissatisfaction scores (BISS) across time point (first trimester, postpartum and at present) and self-reported interoception (MAIA-2).** To assess changes in body satisfaction scores with variations in interoceptive accuracy, a linear mixed model was built (*lme4* package v.1.1–29; [78]). The fixed effects were interoceptive accuracy scores and time point (final trimester, postpartum and current), and an interaction modelled whether the effect of time point on body satisfaction scores depended upon the level of interoceptive accuracy. Participant ID was included as a random effect. Results showed that estimates for retrospective accounts of body satisfaction were lower at the postnatal period than at

**Table 10. Summary of stepwise linear regression predicting maternal engagement in mother-infant stroking across sample of Study 1 and Study 2 ($N = 262$).**

| Predictors | Stepwise regression: Effects of interoceptive focus on engagement in mother-infant stroking | | |
|---|---|---|---|
| | *Estimates* | *CI* | *p* |
| (Intercept) | -0.00 | -0.12 – 0.12 | 1.0000 |
| Self-Regulation | 0.14 | 0.01 – 0.26 | **0.0399** |
| Noticing | 0.13 | 0.00 – 0.26 | **0.0497** |

Observations 262.

$R^2$ / $R^2$ adjusted 0.049 / 0.041

final trimester ($B$ = -8.31, $SE$ = 5.45, $t$ = -1.53). Current body satisfaction scores were also lower than at the postnatal period (-5.34, $SE$ = 6.71, $t$ = -0.985) and compared to the final trimester ($B$ = -13.72, $SE$ = 5.45, $t$ = -2.52) and this effect was statistically significant ($p$ = .013).

The interaction term showed that with increases in interoceptive accuracy, the model estimated an increase in retrospective body satisfaction in the postnatal period, relative to the final trimester ($B$ = 2.84, $SE$ = 7.72, $t$ = 0.37) and this effect was most pronounced for body satisfaction scores at present ($B$ = 11.34, $SE$ = 7.27, $t$ = 1.47) (see Fig 1). A likelihood ratio test indicated that the model including the fixed predictors provided a better fit for the data than a model without them ($\chi^2$ = 19.25; $p$ < .001, $N$ = 52).

**Discussion of Study 2.** The aim of Study 2 was twofold. Firstly, to establish whether the association between measures of self-reported interoceptive focus and stroking could be replicated. Secondly, to establish whether this relationship is also relevant to another dimension of interoception which measures participants' accuracy in detecting heartbeats, i.e., interoceptive accuracy. We replicated the key finding that aspects of self-reported interoceptive bodily focus related to engagement in mother-infant stroking and rocking. Across the combined sample of both studies, the tendency to notice and regulate interoceptive sensations emerged as significant predictors of mothers' engagement in stroking. We found no evidence of a relationship between maternal interoceptive accuracy and engagement in stroking or rocking.

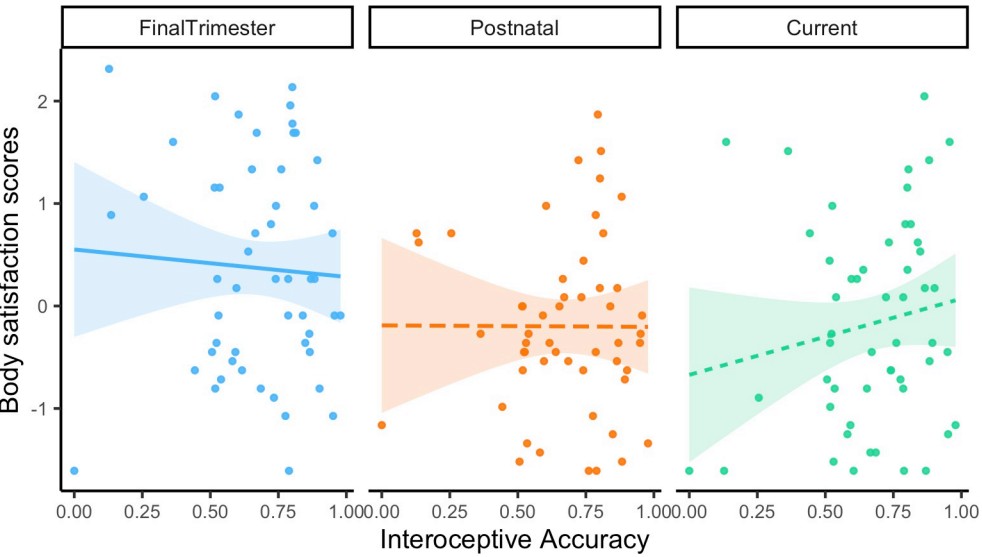

**Fig 1. Relative to the final trimester, body satisfaction at postnatal and at present increases with increases in interoceptive accuracy.**

## General discussion

Across two studies with mothers of a 6- to 18-month-old infant, we examined whether self-reported interoceptive focus related to mothers' engagement in stroking and rocking their infant. In the second study, we extended this to assess whether behavioural performance in a cardiac accuracy task also related to engagement in these caregiving behaviours. These findings are discussed in turn below.

### Interoception and caregiving

Across two hundred and sixty-two participants in our sample, mothers who reported greater tendencies to notice and regulate their own interoceptive states also reported engaging in more stroking with their infant. There may be several possible explanations for this finding. Theoretical accounts suggest that sensitive caregiving requires mothers to first regulate their own emotional experience [30]. Greater focus towards interoceptive states may enable mothers to attune to how their infant's changing arousal states are affecting the dyad and to regulate these states. This is particularly relevant when considering that the *Self-Regulation* dimension describes the ability to regulate distress by sustaining attention to body sensations, e.g., using breath to reduce tension. Such adaptive strategies may enable mothers to recognise how stroking influences their infant's changing arousal states, such as regulating infants' cardio-respiratory rate and body temperature [48, 50]. Our reported effect sizes are small, however, when considering that caregiving is a key feature of infants' early experience, the types of caregiving behaviours infants are exposed to can have important implications for physical and emotional development.

Gentle stroking activates key nodes of the social brain by stimulating C-Tactile fibres that project to the insula [89] and plays an important role in regulating physical arousal when delivered at optimal velocities [59]. It has since been suggested that maternal touch likely contributes to the infant's own developing interoceptive system [1, 2, 90], facilitating the infant's capacity to make predictions about internal and external sensory information [91]. In our samples, engagement in mother-infant rocking and stroking tended to covary. Like stroking, rocking carries a rhythmic quality which affects the degree to which infants are soothed. Rocking optimally soothes infants and promotes sleep when administered at 1.5 Hz/s [92] and respiration has been found to entrain to the rate of rocking [93]. The rhythmic qualities of mother-stroking have also been found to relate to the mother's own arousal [58], as when an infant was calm, mothers with a lower heart rate stroked them more slowly and mothers with a faster heart rate stroked them more quickly. The periodic features of these tactile caregiving behaviours may explain their relevance to interoceptive focus and regulation.

Given that the Holding and Affective Communication factors were not robust for analyses in the current study, further research is warranted to explore whether engagement in these behaviours relates to self-reported interoceptive focus or not. The PICT is useful in assessing mother-infant stroking across daily life. However, the Holding factor on the PICT does not distinguish holding from cuddling, despite previous studies which show that static holding and cuddling have differential effects on infant arousal [51]. Richer measures of caregiving interactions can be captured in naturalistic studies of parent-infant interactions by using wearable video-recording devices [94].

While Study 2 replicated the finding relating mother-infant stroking to self-reported interoceptive focus, we found no relationship to mothers' interoceptive accuracy. Due to the low sample and limitations of collecting heart rate measures online, the null finding should be interpreted with caution. Although estimates of heart rate with the rPPG method have been comparable to estimates from a validated mobile heart rate application in a previous study [87]

and have been utilised in research investigating interoception and social cognition [88], future studies should collect measures of caregiving and interoceptive accuracy in person to test this relationship. Stringent exclusion criteria in the online task meant that the final sample size was small for the intended analysis and the design may have been underpowered for detecting estimated effect sizes, however, the positive relationship observed between interoceptive accuracy and *Attention Regulation* in the MAIA-2 is consistent with previous findings [80] when the heartbeat counting task is used in-person, also lending support for our measure of interoceptive accuracy using the online rPPG method. *Attention Regulation* describes focusing attention to body sensations without being distracted by external stimuli and it is interesting that our findings show *Self-Regulation* and *Attention Regulation* have differential effects with mother-infant stroking and interoceptive accuracy, respectively. As previously mentioned, *Self-Regulation* describes the ability to regulate distress by sustaining attention to body sensations. Mother-infant caregiving likely requires a balance where attention shifts between infants' exteroceptive cues (e.g., crying) and one's own interoceptive cues, thus a tendency to focus wholly on interoceptive vs exteroceptive states may not facilitate mother-infant stroking.

Assuming that the lack of significant relationship between interoceptive accuracy and mother-infant stroking is accurate, the findings suggest that interoceptive accuracy, as measured by the heartbeat counting task, was not related to mothers' reported engagement in stroking behaviours. Importantly, the heartbeat counting task primarily captures a trait-like measure of cardiac accuracy, while certain stressors can cause temporary state-like shifts in cardiac accuracy [95]. For instance, by increasing perturbations to afferent signals, physical exercise is a known moderator of cardiac accuracy [96]. In the context of parent-infant dynamics, infants signal changes in arousal states through distressing and often ambiguous behaviour, i.e., crying. Exposure to own-infant crying vs. unfamiliar infant crying is known to activate brain regions relevant to interoceptive processing, i.e., bilateral insula [97] and over time brain responses to infant cries recruit more regulatory, higher order cortical brain activity (i.e., medial prefrontal cortex) and less threat-related insula activity [98]. Individual differences in state-like changes in interoceptive accuracy may play a more important role in caregiving during high arousal states, as parents monitor how their infant's fluctuating interoceptive states affects their own. Further studies could compare parents' trait accuracy in heartbeat counting task and state-like cardiac accuracy, for instance, inducing a stressor such as infant crying stimuli during a task such as the heartbeat discrimination task [99], a task of interoceptive-exteroceptive switching which more closely simulates the challenge of balancing interoceptive and exteroceptive processing.

## Interoception and body-image

Based on previous evidence that greater body dissatisfaction during pregnancy (i.e., concerns about stretch marks, weight gain, appearance) was associated with less engagement in breast-feeding [68], we explored relationships between stroking and retrospective body satisfaction during pregnancy. We did not find evidence for this, however, retrospective answers may not be reliable. In Study 1, we replicated previous findings of a positive association between maternal self-reported interoceptive focus and pregnancy body satisfaction [67]. It is worth noting that as body (dis)satisfaction likely changes throughout pregnancy, using the BUMPS to measure (retrospective) accounts of pregnancy body satisfaction overall i.e., not specifying a specific trimester or period, was limiting.

In Study 2, self-reported interoceptive focus was moderately associated with greater body satisfaction at present but was not associated with retrospective accounts of body satisfaction during the final trimester of pregnancy, nor the first 12 weeks postpartum. This may be due to

the body satisfaction measurement we used, which may not have captured the nuances of the perinatal and postnatal body that relate to interoception, yet there is no validated measure of postnatal body satisfaction. Across both studies, the retrospective accounts limit the interpretation of the results as women may perceive their body more or less favourably on reflection.

In Study 2, we also predicted that variations in body satisfaction in the final trimester, postnatally and at present would be explained by differences in interoceptive accuracy. Past research found that poorer interoceptive accuracy predicted more body dissatisfaction and self-objectification [61, 89], which may be relevant to pregnancy and the postnatal period, when women experience fluctuating interoceptive sensations and changing body image attitudes. Interestingly, body satisfaction was highest at the final trimester and decreased thereafter, consistent with previous studies [100]. As expected, body satisfaction at present improved with higher interoceptive accuracy, but this effect was not found for retrospective pregnancy and postnatal body satisfaction. As mentioned, the questionnaire measure used and the retrospective aspect may have confounded the results.

Future research should measure interoceptive accuracy and body satisfaction simultaneously across different time periods, to capture how levels of interoception and body satisfaction may change across these periods of substantial physical change. Moreover, other factors may be relevant to interoception, body-image and parent-infant interactional qualities, e.g., type of birth delivery or postnatal depression and anxiety [101], as these experiences may alter how one detects and manages bodily signals after birth.

## Caregiving and body-part vocabulary comprehension

Study 1 explored whether caregiving behaviours played a role in infants' growing understanding of body-part vocabulary, among the earliest words that infants learn [53, 102]. Across 10- to 18-month-olds, infants whose mothers adopted greater frequencies of stroking behaviours overall showed greater comprehension of body-part words. By 3.5 months old, infants show sensitivity to the structural configuration of bodies [103] and hands [104] when presented with scrambled vs intact stimuli, and infants' visual scenes contain more of their own and others' bodies performing actions across development relative to faces [94]. In the action-verb literature, semantic processing of body-related action words, e.g., kick, elicits activity in sensorimotor regions in adults [105, 106] and more recently in toddlers [107]. For body-part nouns, using transcranial magnetic stimulation (TMS) to stimulate arm and leg brain areas has been found to modulate adults' accuracy in processing the words 'arm' and 'leg' [108], supporting embodied accounts of cognition that semantic processing of body-part words involves sensorimotor representation of these states. During episodes of joint attention towards the body in parent-infant interactions, infants are provided with opportunities where parents label body parts and map these, for instance, while tickling and/or singing songs such as 'Heads, Shoulders, Knees and Toes' [109, 110]. Mapping the word reference to the body part through multimodal, i.e., tactile, proprioceptive and motor information may support the representation and recognition of these body parts, and our preliminary results suggest that this may be facilitated in mother-infant stroking, however, further experimental studies on vocabulary comprehension of body-parts are required.**Limitations**

We acknowledge that the scope of the study is limited by using parent-report measures to capture parental engagement in caregiving, which was inevitable given the restraints of the Covid-19 pandemic during which this study took place. Relatedly, it is important to consider that the data were collected during a time when direct and indirect effects (e.g., lockdowns, school closures) on parents' psychological well-being [111] were particularly heightened for mothers, who were more likely to juggle work and care responsibilities [112]. The relation

between parents' own interoceptive abilities i.e., self-regulation, and their caregiving engagement may have been different if examined outside of this context. However, it is also worth noting that the MAIA-2 scores we obtained from our sample are comparable to those of previous studies [14]. Another important consideration is that the affective communication subscale of the PICT could not reliably capture patterns of affective caregiving in our samples, leaving the question of whether these types of behaviours relate to parental interoception unanswered. In future studies, recording parent-infant interactions and measuring their affective communication [113] would enable a more valid measure. Studies that combine parent-report and natural observational measures of parent-infant interactions, i.e., using wearables, will provide more accurate accounts of caregiving as laboratory observations alone are confounded by a lack of ecological validity. Our study does provide insights into the variability in patterns of parent-infant stroking across daily life, which has been under-researched in comparison to studies applying affective touch in experimental settings [114]. Online implementation of the heartbeat counting task in Study 2 resulted in a large proportion of exclusions and subsequent small sample size, presenting challenges for the interpretation of the results which makes it necessary for future studies to address these questions in in-person studies. From a practical perspective, the online heartbeat counting task allows researchers to measure parents' interoceptive accuracy which would be impractical during a laboratory visit when the parent is with their infant at all times, however, the strict exclusion criteria requires that studies initially over-sample which can be difficult when conducting research with special populations such as parents. We hope that the preliminary results present a starting point for investigating whether parental interoceptive processes are involved in caregiving dynamics in early life.

## Conclusion

Our findings contribute to a growing literature on interoceptive involvement in early development and present a starting point for future research. The studies show that maternal engagement in stroking and rocking may be affected by the extent to which mothers focus on and regulate interoceptive states. We cannot comment on the fine-tuned dynamics or appropriateness and efficiency of the caregiving interaction, given the lack of observational data in the studies, but the findings may have important implications given that forms of affectionate touch, e.g., stroking, affect a range of infant outcomes relating to physical and emotional well-being [115]. Further research is warranted to delineate the interoceptive mechanisms involved in caregiving and how maternal interoception plays a role in infants' developing interoception and emotion-regulation across childhood.

## Supporting information

**S1 Checklist. STROBE statement—checklist of items that should be included in reports of observational studies.**
(DOCX)

**S1 File. Confirmatory factor analysis Study 1.**
(DOCX)

**S2 File. Exploratory analyses of stroking-rocking factor.**
(DOCX)

**S3 File. Distribution of body part vocabulary Study 1.**
(DOCX)

**S4 File. Confirmatory factor analysis Study 2.**
(DOCX)

**S5 File. Interoceptive acc and HR correlations Study 2.**
(DOCX)

## Acknowledgments

We would like to thank Matteo Lisi and Joe Bathelt for their guidance on mixed effects models.

## Author Contributions

**Conceptualization:** Rosie Donaghy, Manos Tsakiris.

**Data curation:** Rosie Donaghy.

**Formal analysis:** Rosie Donaghy.

**Funding acquisition:** Manos Tsakiris.

**Investigation:** Rosie Donaghy.

**Methodology:** Rosie Donaghy, Manos Tsakiris.

**Project administration:** Rosie Donaghy, Manos Tsakiris.

**Resources:** Manos Tsakiris.

**Software:** Rosie Donaghy.

**Supervision:** Jeanne Shinskey, Manos Tsakiris.

**Visualization:** Rosie Donaghy.

**Writing – original draft:** Rosie Donaghy.

**Writing – review & editing:** Rosie Donaghy, Jeanne Shinskey, Manos Tsakiris.

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
