## [Decision Letter · Decision Letter 0]

26 Feb 2024

PONE-D-23-19284Maternal interoceptive focus is associated with greater reported engagement in mother-infant stroking and rockingPLOS ONE

Dear Dr. Drysdale,

Thank you for submitting your manuscript to PLOS ONE. Please accept our apologies for the length of the review process; this was because it was extremely difficult to find reviewers to assess it. The reviews are below - these are positive but there are some revisions required. Therefore, we invite you to submit a revised version of the manuscript that addresses the points raised during the review process.

We look forward to receiving your revised manuscript.

Kind regards,

Jane Elizabeth Aspell, PhD

Academic Editor

PLOS ONE

Journal Requirements:

Reviewers' comments:

Reviewer's Responses to Questions

**Comments to the Author**

1. Is the manuscript technically sound, and do the data support the conclusions?

Reviewer #1: Yes

Reviewer #2: Yes

2. Has the statistical analysis been performed appropriately and rigorously? 

Reviewer #1: Yes

Reviewer #2: Yes

3. Have the authors made all data underlying the findings in their manuscript fully available?

Reviewer #1: Yes

Reviewer #2: Yes

4. Is the manuscript presented in an intelligible fashion and written in standard English?

Reviewer #1: Yes

Reviewer #2: Yes

5. Review Comments to the Author

Reviewer #1: Many thanks for the opportunity to review this interesting paper. The report demonstrates the important role that interoception could play in parent-infant caregiving behaviours, and outlines the start of the process of understanding the role of interoception in mother-infant dyadic communication, whilst indicating the flaws of particularly the more objective measures of interoceptive accuracy.

It is clear that the research has been conducted thoroughly, and includes thorough analysis of data and conclusions that largely support the results outlined. I would recommend some adjustments to the manuscript, which I feel could further develop the article.

Major points:

1. Throughout the paper there are issues with the narrative and flow, which make it difficult to follow. There should be a flow throughout of the key hypotheses, from background, to hypotheses, to results then to discussion – some of these are missed at times. This would make the paper easier to follow throughout. Specific examples include:

a. I felt that the explanation of interoception, in relation to the mother-infant relationship was clearly outlined in the introduction, but the narrative relating to body satisfaction and to infant vocabulary was sparce and unclear. For example, an outline or discussion of the competition of cues hypothesis or other explanation why the relationship between exteroception and interoception occurs, would be useful to add context and clarify the relevance of body satisfaction here. This would allow a more holistic explanation of how these concepts fit together. Likewise, there was no mention in the introduction as to why infant vocabulary, and particularly in relation to body parts, might relate to the other variables, yet this was hypothesised. A full explanation of all elements of the hypotheses would have allowed a clearer narrative and flow through the introduction/hypotheses. It may be intuitive to some readers, but explicitly justifying would allow for a clearer narrative in the introduction.

b. From line 334: The subheading is framed as relating to stroking, but the content of that subsection relates to interoception. It therefore feels like a section of the results is missing – relationships between PICT and other variables.

c. From line 600: The results for study 2=two are in a different order to those from study one. It would be easier to follow if either the results were in the same order as the hypotheses, or by order of type of analysis (correlations then regressions).

The results lack acknowledgement of non- significant results. It would be useful to either acknowledge this, or include null results, in the interest of open reporting of data, and to ensure a flowing argument through results and discussion. For example the statement on line 412 “Conversely, engagement in these caregiving behaviours was not associated with pregnancy body satisfaction, nor social touch attitudes.” is outlined as a key outcome in the discussion, but was not outlined in statements in the results – it appeared in the table, but as it is framed as a key result in the discussion it might be useful to overtly outline it in the results section. This detracts from the ability to follow the argument through.

Minor points:

Introduction:

1. Line 65-66: It is unclear how people’s interoceptive insights influence how they interact with users. It might help if this could be made more explicit

2. Line 79: Further examples of physiological reactions could further the readers understanding of the impact of infant distress, for example milk let down.

3. Line 124: Your measures are of body (dis)satisfaction, but the introduction refers to the concept of body image. There are subtle differences between these concepts which I feel could be clarified and discussed, possibly with key definitions and distinctions of both, or just referring to body satisfaction throughout.

Method:

4. Line 159: The explanation of the power analysis was clear, but the justification for using a small effect size is missing. Likewise, the process of the power analysis for study 2 is not clear? Presumably if this was the same as for study one, the sample size fell short of this, in which case an explanation/acknowlegement for this would be helpful.

5. Line 172: Incomplete responses were mentioned but it is not clear if the research team excluded the participants for single data points, or just for whole measures that were missing. The paper would benefit from this being clarified.

6. Line 247: BUMPS is a 19 item measure, but here it is described as 20 items. Further to this, it is unclear whether participants were asked about a specific time point in pregnancy when they were recalling retrospectively - Could be clarified, and an acknowledgement included that body (dis)satisfaction changes during pregnancy.

7. Line 439: The sample appears to be varied, across 3 countries. It would be useful to acknowledge a quick explanation of why those countries were chosen, to possibly pick this up in discussion, in terms of the potential drawbacks of this, e.g., maternity leave differences in UK/USA might cause behavioural differences from mother towards infant.

Results:

8. Table 3 and table 8: Could you clarify what the last column refers to – could it be described as ‘possible range’ rather than ‘range’?

9. Line 370 and 456: It would be useful to explain why those particular subscales of the MAIA have been chosen, as it is unclear in the manuscript.

10. Line 482-483: The paragraph starts: “Before the task, participants had to successfully answer questions regarding these requirements to proceed to the experiment, e.g., “I can complete the study on a mobile”, however the requirements referred to here have not been explained prior to this, but it is written as if it is referring to the previous paragraph, which it is not.

11. Line 494: Sentence “Heart rate estimation was conducted offline using the video recordings” is not clear – can you clarify what is this referring to? Is it what the participants did, or some calculation conducted by the researcher?

12. Line 496: Can more explanation be given about the processes relating to the interoceptive accuracy task, the instructions etc. This could be followed up in the results with a descriptive summary of their estimations.

13. Line 533-539: It would be useful to clarify which time point the alpha for BISS came from, or to give one for each timepoint?

14. Line 638: Was there an analysis comparing current vs postnatal to add in here, so that there are comparisons of all 3 timeframes?

15. Line 718: Attention regulation (construct captured by the subscale from the MAIA) has been explained, but it would be useful to explain self-regulation too so that the subtle differences between them are clearer.

Discussion:

16. Limitations: It would be helpful to acknowledge the small sample size and large proportion of exclusions in study 2 here. Also, the issues of relying on maternal reporting of mother-infant communication – the skewness was mentioned earlier in the report, but not addressed here. Possibly therefore suggesting watching videos or live-video conferencing of dyadic interactions between mother and infant.

Reviewer #2: This is an interesting paper on an important topic; and I applaud the authors efforts to conduct this type of much needed research.

I have two minor comments:

a) the Discussion section (the general) tends to repeat the Discussion 1 and 2. I suggest to present the studies one after the other, and put all the discussion at the end. I find many § repetitive

b) I wonder how much of the patterns you found are related precisely to the fact that the study was conducted during a stressful period fo everyone. I believe this needs to be clearly discussed in the paper, not just briefly mentioned in the limitations.

6. PLOS authors have the option to publish the peer review history of their article (what does this mean?). If published, this will include your full peer review and any attached files.

Reviewer #1: No

Reviewer #2: No

---

## [Author Response · Author response to Decision Letter 0]

11 Mar 2024

Academic Editor:

Thank you for submitting your manuscript to PLOS ONE. Please accept our apologies for the length of the review process; this was because it was extremely difficult to find reviewers to assess it. The reviews are below - these are positive but there are some revisions required. Therefore, we invite you to submit a revised version of the manuscript that addresses the points raised during the review process.

REPLY: Thank you for handling our manuscript. Below we provide a point-by-point reply to all the comments from the reviewers, and we hope that these have addressed all the issues. Our revisions to the manuscript are in red font (in “Revised with track changes.docx”). 

Review Comments to the Author:

Reviewer #1: Many thanks for the opportunity to review this interesting paper. The report demonstrates the important role that interoception could play in parent-infant caregiving behaviours, and outlines the start of the process of understanding the role of interoception in mother-infant dyadic communication, whilst indicating the flaws of particularly the more objective measures of interoceptive accuracy.

It is clear that the research has been conducted thoroughly, and includes thorough analysis of data and conclusions that largely support the results outlined. I would recommend some adjustments to the manuscript, which I feel could further develop the article.

REPLY: Thank you for reviewing our manuscript. We hope that the replies provided below and the changes in the manuscript address all your comments. 

Major points:

1. Throughout the paper there are issues with the narrative and flow, which make it difficult to follow. There should be a flow throughout of the key hypotheses, from background, to hypotheses, to results then to discussion – some of these are missed at times. This would make the paper easier to follow throughout. Specific examples include:

a. I felt that the explanation of interoception, in relation to the mother-infant relationship was clearly outlined in the introduction, but the narrative relating to body satisfaction and to infant vocabulary was sparce and unclear. For example, an outline or discussion of the competition of cues hypothesis or other explanation why the relationship between exteroception and interoception occurs, would be useful to add context and clarify the relevance of body satisfaction here. This would allow a more holistic explanation of how these concepts fit together. 

REPLY: Thank you for this suggestion, we have (after defining body image) alluded to the relevant literature suggesting the inverse relation between external and internal representation of the body (e.g., Ainley et al., 2012; Emmanuelsen et al., 2015; Eshkevari et al). We have also made sure to explicitly reason as to why this may be relevant to pregnancy/postpartum and with links to caregiving. We now write on page 6, starting from line 128:

Moreover, evidence suggests that interoception plays a key role in shaping body image (61–63), defined as the conscious, mostly visual, mental representation of an individual’s body appearance, including its perceptual aspects (e.g., its shape, size) and its affective-cognitive evaluation (e.g., feeling dis/satisfied with the appearance or size of the body) (64). For individuals with have poorer interoceptive abilities, their body representation seems to be more heavily influences by exteroceptive (i.e., visual) features of their body, which may also explain why they are more susceptible to body-image disturbances (65). Accordingly, lower interoceptive accuracy is associated with greater body dissatisfaction (66). The substantial bodily changes experienced during pregnancy and the postpartum suggest these may be sensitive periods for interoception and body image, but few studies have investigated this. In a sample of pregnant women, lower self-reported interoceptive focus was associated with greater body dissatisfaction during pregnancy (67), which has been linked to less engagement in breastfeeding (68), highlighting that interoception and body image have the potential to influence parent-infant caregiving. 

Likewise, there was no mention in the introduction as to why infant vocabulary, and particularly in relation to body parts, might relate to the other variables, yet this was hypothesised. A full explanation of all elements of the hypotheses would have allowed a clearer narrative and flow through the introduction/hypotheses. It may be intuitive to some readers, but explicitly justifying would allow for a clearer narrative in the introduction.

REPLY: We have included a rationale in the introduction relating caregiving to body part comprehension. We have cited a relevant paper highlighting a role for parents’ verbal communication of interoceptive states (i.e., MacCormack et al, 2020) which fits in with the flow of the section, because it touches on how parental interoception may support infants’ awareness of bodily states through caregiving. We now write, starting from line 107:

In childhood, evidence suggests that the way parents verbally communicate their interoceptive states may shape how children identify and appraise their own bodily sensations (52). Given that the words for body parts are among the first words infants learn (53), the caregiving environment may shape children’s identification and labelling of their body and its parts even earlier. 

b. From line 334: The subheading is framed as relating to stroking, but the content of that subsection relates to interoception. It therefore feels like a section of the results is missing – relationships between PICT and other variables.

REPLY: We have swapped the wording to make this subheading more accurate. The subheading is now “Associations between maternal self-reported interoception and mother-infant stroking, pregnancy body satisfaction and social touch attitudes”. We also explicitly stated the non-significant result between stroking (PICT) and retrospective pregnancy body satisfaction (BUMPS) in that section (which also links to your point d. below). 

c. From line 600: The results for study 2=two are in a different order to those from study one. It would be easier to follow if either the results were in the same order as the hypotheses, or by order of type of analysis (correlations then regressions).

REPLY: Thanks to the Reviewer’s comments we have now improved the presentation of our results and statistical analysis in Study 2, to match those of Study 1; integrating the correlations together before the regressions as suggested (which links to point d. below). We agree that this is better for consistency and cross referencing between the results of the studies. Thank you for this suggestion. 

d.The results lack acknowledgement of non- significant results. It would be useful to either acknowledge this, or include null results, in the interest of open reporting of data, and to ensure a flowing argument through results and discussion. For example the statement on line 412 “Conversely, engagement in these caregiving behaviours was not associated with pregnancy body satisfaction, nor social touch attitudes.” is outlined as a key outcome in the discussion, but was not outlined in statements in the results – it appeared in the table, but as it is framed as a key result in the discussion it might be useful to overtly outline it in the results section. This detracts from the ability to follow the argument through.

REPLY: We have explicitly stated the non-significant results i.e., between the subscales of the maia and retrospective body satisfaction at final trimester and postpartum, in the section with the correlations (before the regression). We have also added the contrast between current vs postnatal in the mixed model (as per your point 14 below). 

Minor points:

Introduction:

1. Line 65-66: It is unclear how people’s interoceptive insights influence how they interact with users. It might help if this could be made more explicit

REPLY: Yes we see how this was unclear and sparse. We have cleared this up by removing that phrasing and adding citations to highlight its link to emotion regulation, in the following paragraph (Dedentado et al., 2023; Zamariola et al., 2019). We now write on page 3, starting line 68:

Poorer self-reported interoceptive abilities (as reported in the MAIA-2) have been associated with deficits in emotion regulation in healthy populations (20) which contributed to depressive symptomatology (21). 

2. Line 79: Further examples of physiological reactions could further the readers understanding of the impact of infant distress, for example milk let down.

REPLY: Thank you for this suggestion, we have added the example of stimulating milk flow from the breast, citing Bornstein et al (2017). 

3. Line 124: Your measures are of body (dis)satisfaction, but the introduction refers to the concept of body image. There are subtle differences between these concepts which I feel could be clarified and discussed, possibly with key definitions and distinctions of both, or just referring to body satisfaction throughout.

REPLY: To resolve this, we have added a brief definition of body image and its relation to body satisfaction (line 124), and more explicitly stated when findings of previous research measured body satisfaction (i.e., rather than using body image as a broad term). For instance, on lines 749 and 750 we have reworded ‘body image concerns’ to ‘body dissatisfaction’ and explicitly outlined what quantified this in that study’s measure e.g., stretch marks, appearance and weight gain. We agree that this should provide more clarity for the reader. 

Method:

4. Line 159: The explanation of the power analysis was clear, but the justification for using a small effect size is missing. Likewise, the process of the power analysis for study 2 is not clear? Presumably if this was the same as for study one, the sample size fell short of this, in which case an explanation/acknowlegement for this would be helpful.

REPLY: Yes, we attempted to recruit the same number of participants as Study 1. We have acknowledged that our sample size fell short due to difficulties in recruiting because of the stricter inclusion criteria (i.e., participants needed to complete the study on a laptop with a webcam). 

5. Line 172: Incomplete responses were mentioned but it is not clear if the research team excluded the participants for single data points, or just for whole measures that were missing. The paper would benefit from this being clarified.

REPLY: We have clarified that these were removed because they did not finish the online survey and therefore their responses were not included (lines 178-179). 

6. Line 247: BUMPS is a 19 item measure, but here it is described as 20 items. 

REPLY: Thank you for correcting us on this, we have rectified this in the paper (when referring to it as a 19-item questionnaire and the possible range of scores in the descriptive statistics). We also noticed in the process that we had labelled the max scores in the reporting of the interoception subscales in the descriptives, rather than the means, so we have rectified this too. 

Further to this, it is unclear whether participants were asked about a specific time point in pregnancy when they were recalling retrospectively - Could be clarified, and an acknowledgement included that body (dis)satisfaction changes during pregnancy.

REPLY: Thank you for the suggestion, in lines 254-259 we have clarified this by explicitly stating that in the original study women responded across all stages of pregnancy (i.e., first through to third trimester), and that in our study we did not ask about a specific time during pregnancy. Later (lines 767-770) we have acknowledged that, as you describe, body (dis)satisfaction changes during pregnancy and this coarse measure of an ‘overall’ pregnancy body satisfaction would not have captured these changes. To the latter point, we now write, starting page 37 (“Interoception and body-image” section): 

It is worth noting that as body (dis)satisfaction likely changes throughout pregnancy, using the BUMPS to measure (retrospective) accounts of pregnancy body satisfaction overall i.e., not specifying a specific trimester or period, was limiting.

7. Line 439: The sample appears to be varied, across 3 countries. It would be useful to acknowledge a quick explanation of why those countries were chosen, to possibly pick this up in discussion, in terms of the potential drawbacks of this, e.g., maternity leave differences in UK/USA might cause behavioural differences from mother towards infant.

REPLY: 

Thank you for raising this point. For Study 2, we had more inclusion criteria (e.g. “female participants between 18 and 40 years old, with a child born in the year 2020, and with access to a computer or laptop, and consent to be video recorded. There were too few eligible participants on Prolific who met these criteria and we therefore opened up recruitment to other countries with native English speakers, and raising the number of eligible participants to n=380. In that sense, the recruitment from these countries was not theoretically motivated. We ended up with n= 75 from UK, n=2 from Ireland, and n=34 from USA. While there are differences in maternity leave entitlements across these countries and these may have affected, as the Reviewer suggests, interaction between mothers and infants in the first few years of life, we do not have data on whether and how our participants made use of their maternity leave entitlements. The potential across-countries differences on maternity leave and other factors could play a role on mother-infant interactions, as you suggest, but we do not have any data to speak to that point, however we agree that this is an important research question for future studies that can focus on cross-cultural differences in interoceptive development.

Results:

8. Table 3 and table 8: Could you clarify what the last column refers to – could it be described as ‘possible range’ rather than ‘range’?

REPLY: Yes, this is the possible range and we have reworded this accordingly.

9. Line 370 and 456: It would be useful to explain why those particular subscales of the MAIA have been chosen, as it is unclear in the manuscript.

REPLY: We have explicitly stated our selection of the MAIA subscales of interest from study 1.

10. Line 482-483: The paragraph starts: “Before the task, participants had to successfully answer questions regarding these requirements to proceed to the experiment, e.g., “I can complete the study on a mobile”, however the requirements referred to here have not been explained prior to this, but it is written as if it is referring to the previous paragraph, which it is not.

REPLY: Thank you for spotting this. We have resolved this in relation to your next comments (point 11 and 12) by rewording this section and providing more detail. We hope this is clearer. 

11. Line 494: Sentence “Heart rate estimation was conducted offline using the video recordings” is not clear – can you clarify what is this referring to? Is it what the participants did, or some calculation conducted by the researcher?

REPLY: Thank you for your suggestion, we initially removed some of the details for brevity but having re-read it we agree that it requires further explanation. We have added further details about the heart rate estimation procedure (and the task as requested below). We have tried to keep this as concise as possible while providing the necessary details to assist the reader's understanding. We now write, starting on page 24 (“Online heartbeat counting task: Remote photoplethysmography” section):

Participants completed a measure of interoceptive accuracy that could be obtained remotely. Specifically, the Heartbeat Counting Task (HCT) (13) was integrated with a novel method for calculating heart rate from pixel changes in video recordings of human skin (84). Remote photoplethysmography (rPPG) uses a light source to measure the variation in oxygenated blood flow and corresponds to phasic changes in heart rate (85). rPPG works via photo-amplification, which detects variations in the reflected colours of the skin caused by changes in capillary tissue movement that c

---

## [Decision Letter · Decision Letter 1]

12 Apr 2024

Maternal interoceptive focus is associated with greater reported engagement in mother-infant stroking and rocking

PONE-D-23-19284R1

Dear Dr. Donaghy,

We’re pleased to inform you that your manuscript has been judged scientifically suitable for publication and will be formally accepted for publication once it meets all outstanding technical requirements.

Kind regards,

Jane Elizabeth Aspell, PhD

Academic Editor

PLOS ONE

Reviewers' comments:

Reviewer's Responses to Questions

**Comments to the Author**

1. If the authors have adequately addressed your comments raised in a previous round of review and you feel that this manuscript is now acceptable for publication, you may indicate that here to bypass the “Comments to the Author” section, enter your conflict of interest statement in the “Confidential to Editor” section, and submit your "Accept" recommendation.

Reviewer #1: All comments have been addressed

2. Is the manuscript technically sound, and do the data support the conclusions?

Reviewer #1: Yes

3. Has the statistical analysis been performed appropriately and rigorously? 

Reviewer #1: Yes

4. Have the authors made all data underlying the findings in their manuscript fully available?

Reviewer #1: Yes

5. Is the manuscript presented in an intelligible fashion and written in standard English?

Reviewer #1: Yes

6. Review Comments to the Author

Reviewer #1: The authors have addressed all the comments and I feel like the manuscript has improved with these and their other adjustments

7. PLOS authors have the option to publish the peer review history of their article (what does this mean?). If published, this will include your full peer review and any attached files.

Reviewer #1: No

---

## [Editor Report · Acceptance letter]

29 Apr 2024

PONE-D-23-19284R1 

PLOS ONE

Dear Dr. Donaghy, 

I'm pleased to inform you that your manuscript has been deemed suitable for publication in PLOS ONE. Congratulations! Your manuscript is now being handed over to our production team.

Kind regards, 

on behalf of

Dr. Jane Elizabeth Aspell 

Academic Editor

PLOS ONE